# Intra- and inter-molecular regulation by intrinsically-disordered regions governs PUF protein RNA binding

Chen Qiu[1], Zihan Zhang [2], Robert N. Wine[1], Zachary T. Campbell [3], Jun Zhang [2] & Traci M. Tanaka Hall [1] ✉

PUF proteins are characterized by globular RNA-binding domains. They also interact with partner proteins that modulate their RNA-binding activities. *Caenorhabditis elegans* PUF protein *fem-3* binding factor-2 (FBF-2) partners with intrinsically disordered Lateral Signaling Target-1 (LST-1) to regulate target mRNAs in germline stem cells. Here, we report that an intrinsically disordered region (IDR) at the C-terminus of FBF-2 autoinhibits its RNA-binding affinity by increasing the off rate for RNA binding. Moreover, the FBF-2 C-terminal region interacts with its globular RNA-binding domain at the same site where LST-1 binds. This intramolecular interaction restrains an electronegative cluster of amino acid residues near the 5′ end of the bound RNA to inhibit RNA binding. LST-1 binding in place of the FBF-2 C-terminus therefore releases autoinhibition and increases RNA-binding affinity. This regulatory mechanism, driven by IDRs, provides a biochemical and biophysical explanation for the interdependence of FBF-2 and LST-1 in germline stem cell self-renewal.

Intrinsically disordered regions (IDRs) are prevalent in eukaryotic proteomes[1]. Despite the lack of well-defined structures, they engage in molecular recognition and molecular assembly and play important roles in cellular signaling and regulation[2,3]. RNA-binding proteins are enriched with IDRs[4]. IDRs can influence RNA-binding activities and promote formation of higher-order ribonucleoprotein complexes, such as membrane-less granules[5–10]

Proper control of mRNA localization, translation, and stability requires recognition of sequence or structural motifs by RNA-binding proteins to select specific target mRNAs[11–13]. RNA recognition can be modulated by physical interaction of RNA-binding proteins with partner proteins via globular domains or IDRs[14,15]. These collaborative RNA recognition complexes produce distinct regulatory capabilities not seen with the individual proteins[14–18]. Understanding the mechanistic details and the impact that partner proteins have on RNA-binding proteins is critical for deciphering the cellular functions of the complexes. Here we use the *C. elegans* PUF protein FBF-2 (*fem-3* Binding Factor-2) and its partner protein LST-1 (Lateral Signaling Target-1) as model RNA regulatory proteins to explore the molecular mechanism underlying their collaboration, focusing on the roles of their IDRs (Fig. 1a, b). These two proteins and their partnership are crucial for repression of *gld-1* translation to maintain germline stem cells (GSCs)[19–24]. A mechanistic understanding of the LST-1–FBF interaction is therefore of particular importance.

FBF-2 has an architecture characteristic of all PUF proteins, including Drosophila Pumilio and human PUM1 and PUM2[25–31]. The signature PUF RNA-binding domain (RBD), also called the Pumilio-Homology Domain (PUM-HD), is flanked by extended N-terminal and short C-terminal IDRs (Fig. 1a). The PUF RBD with eight α-helical repeats recognizes RNA sequence elements[32–34]. The RBD of the prototypical human Pumilio1 recognizes an 8-nt RNA sequence (5′-UGUAnAUA, n=any nucleotide) with each repeat binding to one

[1]Epigenetics and Stem Cell Biology Laboratory, National Institute of Environmental Health Sciences, National Institutes of Health, Research Triangle Park, NC, USA. [2]Department of Chemistry, University of Alabama at Birmingham, Birmingham, AL, USA. [3]Department of Anesthesiology, University of Wisconsin School of Medicine and Public Health, Madison, WI, USA. ✉e-mail: hall4@niehs.nih.gov

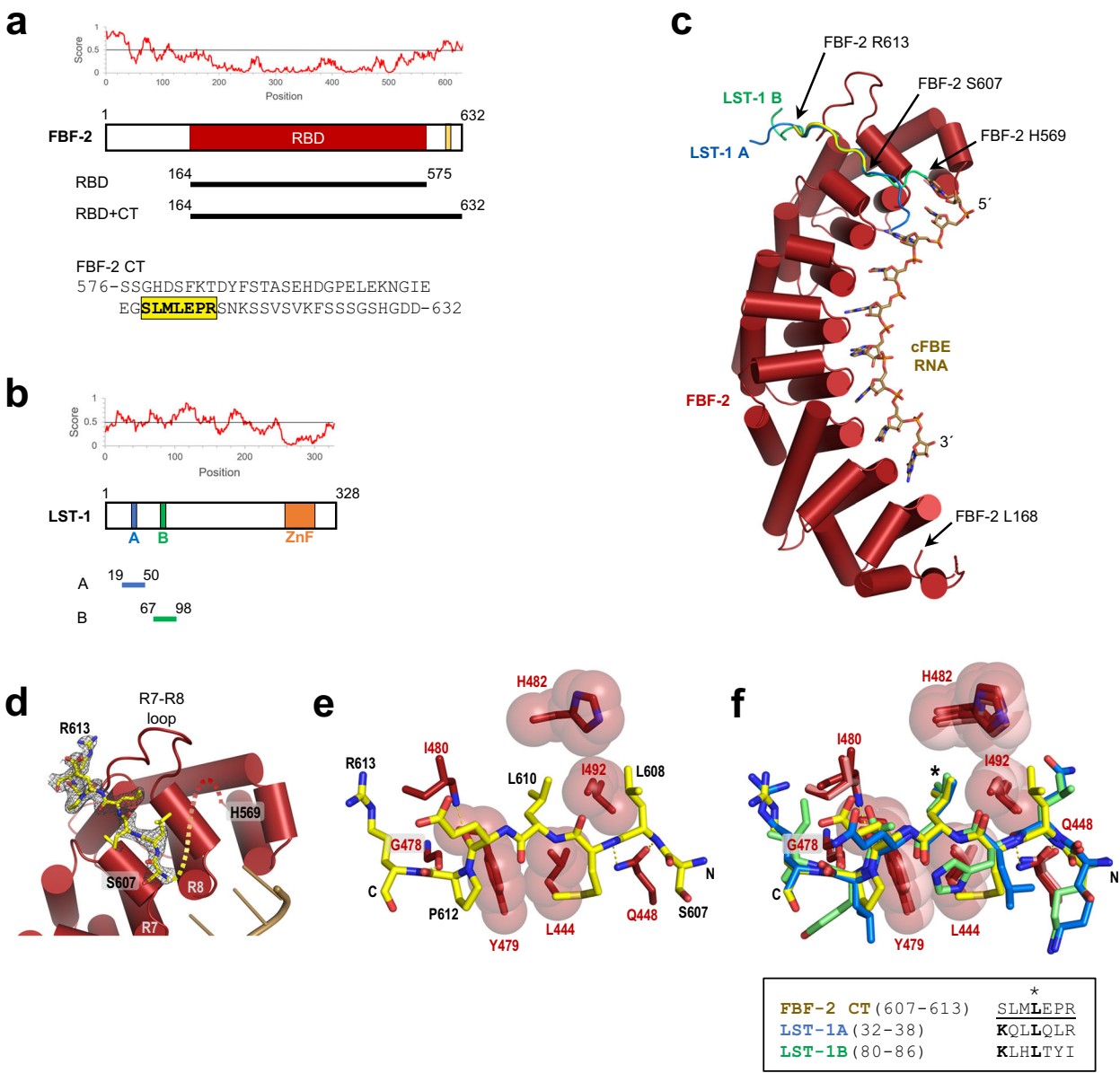

**Fig. 1 | The FBF-2 C-terminal tail binds to its RBD like LST-1 A and B.** Intrinsically disordered regions in **a** FBF-2 and **b** its partner protein LST-1. Schematic drawings of FBF-2 and LST-1 are shown with predicted disorder plot from IUPred2A[70,71] above and protein constructs used below. The scores in the predicted disorder plot indicate the probability of the residue being disordered. The FBF-2 CT PIM (residues 607-613) is indicated by a yellow box. The FBF-2 CT sequence is shown with its PIM highlighted yellow. FBF-2 contains a structured RBD, and LST-1 contains a predicted zinc finger (ZnF). The N-terminal region of LST-1 is sufficient for GSC self-renewal, and the C-terminal region provides feedback to limit GSC number[72]. **c** Crystal structure of FBF-2 RBD + CT. A ribbon diagram of the FBF-2 RBD with cylindrical helices (red) is shown in complex with cFBE RNA (tan stick model) and its CT peptide (yellow). LST-1 A (blue) and LST-1 B (green) are superimposed.

**d** Composite 2F$_o$-F$_c$ omit map contoured at 1 σ is superimposed on the crystal structure of FBF-2 RBD + CT in complex with cFBE RNA. The CT is shown as a yellow stick model. **e** FBF-2 RBD interacts with L610 in its CT. The FBF-2 CT peptide (yellow stick model) is shown. FBF-2 RBD residues that interact with the CT are displayed as stick models. Side chains that form a hydrophobic pocket for L610 are shown with transparent spheres. Hydrogen bond interactions between the FBF-2 RBD and backbone atoms of its CT are indicated with dashed lines. **f** FBF-2 interacts with core sequences in its CT, LST-1 A, and LST-1 B. Crystal structures of FBF-2 RBD + CT, FBF-2 RBD in complex with LST-1 A and LST-1 B are superimposed. The LST-1 peptides are shown as stick models (LST-1 A, blue; LST-1 B, green). FBF-2 RBD residues that interact with the peptides are shown as in **e**. The critical leucine residues are indicated with an asterisk.

nucleotide[33]. *C. elegans* FBF-1 and FBF-2 (referred to collectively as FBF) are more flexible in their RNA sequence recognition[35–37]. A well-studied FBF-binding element (FBE) in the *gld-1* 3′ untranslated region, called FBEa, has a 9-nt core sequence beginning with UGU and is preceded by an upstream cytosine that increases RNA-binding affinity, 5′-CAU-GUGCCAUA-3′[19,38–40]. More recently, an 8-nt compact FBE (cFBE) was identified, 5′-CUGUGAA(A/U)n-3′[17,36]. Although FBF-2 is sufficient to recognize *gld-1*, its interaction with LST-1 is required for GSC self-renewal[21,22].

LST-1 is expressed in GSCs in response to signaling from the stem cell niche[20]. The LST-1 protein is largely unstructured (Fig. 1b), including an N-terminal region that is sufficient for self-renewal of GSCs[22]. This LST-1 IDR region harbors two PUF Interacting Motifs (PIMs), LST-1 A and LST-1 B, that interact with FBF-2, each with a KxxL sequence[17,22]. Crystal structures of FBF-2 bound to LST-1 A or B identify a seven-residue core region, including the KxxL sequence, that binds to FBF-2 with similar peptide backbone structure[17,18]. The two LST-1 PIMs are biologically redundant for GSC maintenance in

nematodes[22], but they have distinct FBF-2-binding affinities and distinct effects on FBF-2 RNA-binding affinity in vitro. An LST-1 B peptide (LST-1[67–98]) binds tightly to FBF-2 and decreases RNA-binding affinity of its RBD, while an LST-1 A peptide (LST-1[19–50]) binds with lower affinity to FBF-2 and has no effect on RNA-binding affinity[17,18].

Here we investigate the mechanism by which LST-1 and FBF-2 interact to control *gld-1* RNA. We report a 2.1-Å crystal structure of FBF-2 in complex with RNA that identifies a PIM within the intrinsically disordered FBF-2 C-terminal tail (CT) bound to the same site on the RBD as LST-1. When comparing RNA binding by the FBF-2 RBD to RNA binding by the RBD plus the CT (RBD + CT), we discover that the CT is autoinhibitory. Addition of the CT to the RBD decreases RNA-binding affinity by increasing the off-rate. We demonstrate that interaction between the FBF-2 CT PIM and RBD is required for autoinhibition. This interaction between the CT PIM and RBD appears to restrain an electronegative cluster near the 5′ end of the bound RNA. Consistent with this notion, alanine substitutions in the electronegative cluster diminish autoinhibition. LST-1 A and B increase the RNA-binding affinity of FBF-2 RBD + CT, and we find that LST-1 A and LST-1 B bind more tightly to the RBD than the CT. Together, these findings suggest that LST-1 enhances FBF-2 RNA-binding affinity by displacing its CT, thereby alleviating autoinhibition. This regulatory mechanism, driven by IDRs, provides a biochemical and biophysical explanation for the interdependence of FBF-2 and LST-1 in GSC self-renewal.

## Results

### FBF-2 CT binds FBF-2 RBD on the same surface as LST-1
Our previous structural and biochemical studies relied on FBF-2[164–575], a fragment with the RBD that lacks its intrinsically disordered C-terminal tail but retains its in vivo RNA-binding specificity (Fig. 1a)[17,18,36,39,40]. The FBF-2 CT has been shown to be functionally important[41]. A worm strain that expresses truncated FBF-2 lacking the C-terminal 26 residues (607-632) was generated in an *fbf-1* loss-of-function background. Approximately 11% of the mutant progeny were sterile due to failure to initiate oogenesis, indicating a germline defect. In contrast, worms expressing full-length FBF-2 in an *fbf-1* loss-of-function background were 0% sterile. To explore possible roles of the FBF-2 CT, we expressed and purified an FBF-2 fragment containing both RBD and CT (FBF-2[164–632]). This FBF-2 RBD + CT had increased thermal stability compared to the RBD alone (Supplementary Table 1). We determined a crystal structure of FBF-2 RBD + CT in complex with a cFBE RNA (5′-CUGUGAAUG-3′) at 2.1 Å resolution (Supplementary Table 2, Fig. 1c). The electron density map revealed new density, not present in the previous RBD structures. This additional density appeared near a loop in the RBD that connects repeats 7 and 8 (Fig. 1d). This R7-R8 loop (residues 476-489) also binds to both LST-1 A and LST-1 B peptides in crystal structures of FBF-2 RBD in complex with LST-1 (Fig. 1c)[17,18]. The sequence of FBF-2 CT residues 607-613 matched this new electron density, but other C-terminal residues (aa 570-606 and 614-632) were disordered.

In the FBF-2 RBD + CT/cFBE crystal structure, the seven visible FBF-2 CT residues (SLMLEPR[607–613]) align structurally with the core regions of LST-1 A (residues 32-38) and LST-1 B (residues 80-86) and made remarkably similar contacts to the RBD (Fig. 1e, f). FBF-2 L610 occupied the same position as the key leucines, L35 and L83, in the LST-1 A and B PIMs, and like LST-1 A L35 and LST-1 B L83, FBF-2 L610 was seated in a hydrophobic binding pocket formed by the Y479, L444, I492, and H482 residues of FBF-2 (Fig. 1e, f). The Cα backbone atoms of the FBF-2 CT also superimposed well with those of core residues 32-38 of the LST-1 A peptide and core residues 80-86 of the LST-1 B peptide. FBF-2 RBD residues Q448, G478, and I480 contacted the peptide backbone atoms of the FBF-2 CT, LST-1 A, and LST-1 B. Thus, the FBF-2 CT contains an intrinsic PIM.

### FBF-2 CT and partner proteins compete for binding to the RBD
Given that the core residues of three distinct PIMs−FBF-2 CT, LST-1 A, and LST-1 B−all bind to the same site of FBF-2, we hypothesized that the intramolecular FBF-2 CT binding could compete with LST-1 A and LST-1 B binding. To assess that idea, we investigated how the CT affects the protein interaction affinities of the FBF-2 RBD to LST-1 A and LST-1 B, using isothermal titration calorimetry (ITC, Supplementary Table 3). No RNA was included in these experiments. We first measured the binding affinity of an isolated CT peptide and the FBF-2 RBD (no CT), by titrating the peptide, FBF-2[601–632], into an ITC cell containing the RBD. The CT peptide bound to the RBD with weaker affinity ($K_d = 35\,\mu M$, Fig. 2a) than measured previously for LST-1 A ($K_d = 2.1\,\mu M$) or LST-1 B ($K_d = 0.05\,\mu M$, Fig. 2b)[18]. That lower affinity of the CT peptide is consistent with crystal structures showing that it makes fewer contacts with the RBD than we previously observed for LST-1 A or B[17,18].

We next compared the protein interaction affinities of LST-1 A or LST-1 B with either FBF-2 RBD + CT or RBD alone (Fig. 2c−g, Supplementary Table 3). For these experiments, we titrated the LST-1 peptide into an ITC cell containing either RBD + CT or RBD. We found that the FBF-2 CT greatly reduced FBF-2 affinity for LST-1 A and LST-1 B. Binding of LST-1 A to FBF-2 RBD + CT was too weak to be detected by ITC (Fig. 2c) and thus considerably weaker than the affinity of LST-1 A to the RBD alone ($K_d = 2.1\,\mu M$)[18]. The binding of LST-1 B to RBD + CT was ~50-fold weaker than to the RBD alone ($K_d = 2.7\,\mu M$ for RBD + CT vs 0.05 μM for RBD, Fig. 2b, d)[18]. The lower protein interaction affinities for LST-1 A and LST-1 B caused by the FBF-2 CT suggested that the CT and LST-1 compete for binding to the RBD. To examine this idea, we next asked if the reduced binding affinity for LST-1 relies on the CT interaction with RBD. L610 in the FBF-2 CT is equivalent to the key leucines in LST-1 PIMs. Mutation of the key leucines in LST-1 disrupt interaction with FBF-2[17], and the L610A mutation was designed to test the importance of the interaction between the RBD and CT. We confirmed that the L610A mutation disrupts interaction with the FBF-2 RBD by ITC (Fig. 2e, Supplementary Table 3). The L610A substitution decreased the thermal stability of FBF-2 RBD + CT with a melting temperature of the mutated protein near that of the RBD alone (Supplementary Table 1), suggesting that interaction of the CT with the RBD is stabilizing. We found that FBF-2 RBD + CT L610A bound to LST-1 A and B with affinities similar to the FBF-2 RBD alone ($K_d = 2.4\,\mu M$ for LST-1 A and $K_d = 0.06\,\mu M$ for LST-1 B, Fig. 2b, f, g). Therefore, the interaction between the FBF-2 CT and RBD via L610 is critical for the reduced affinity of RBD + CT for LST-1 A and B, apparently by competing with LST-1 interaction.

### FBF-2 CT autoinhibits RNA-binding affinity
Based on our previous observation that LST-1 B weakens the RNA-binding affinity of FBF-2 RBD[17,18], we hypothesized that the FBF-2 CT might similarly modulate FBF-2 RNA-binding affinity. To ask if the CT alters FBF-2's RNA binding, we compared RNA-binding affinities of FBF-2 RBD and FBF-2 RBD + CT. Specifically, we measured their binding affinities to *gld-1* RNA with its FBEa binding element, using electrophoretic mobility shift assays (EMSAs) (Fig. 3a, b; Supplementary Table 4). Indeed, FBF-2 RBD + CT had a nearly 5-fold weaker affinity for *gld-1* FBEa RNA than FBF-2 RBD alone ($K_d = 334\,nM$ for FBF-2 RBD + CT and 70 nM for RBD alone). This substantial decrease is greater than the effect of LST-1 B on RBD ($K_d = 176\,nM$, Supplementary Fig. 1a), and the CT effect is intramolecular.

To confirm that the reduced RNA-binding affinity was due to the CT interacting with the RBD, we also measured RNA-binding affinity of FBF-2 RBD + CT proteins mutated in the R7-R8 loop (Y479A) or in the C-terminal tail (L610A). Y479 is a critical part of the hydrophobic pocket that binds to key leucine residues in partner proteins (Fig. 1f), and its mutation to an alanine (Y479A) disrupts interaction between FBF-2 and LST-1[17]. Like L610A, the Y479A substitution decreased the thermal stability of FBF-2 RBD + CT (Supplementary Table 1). Both

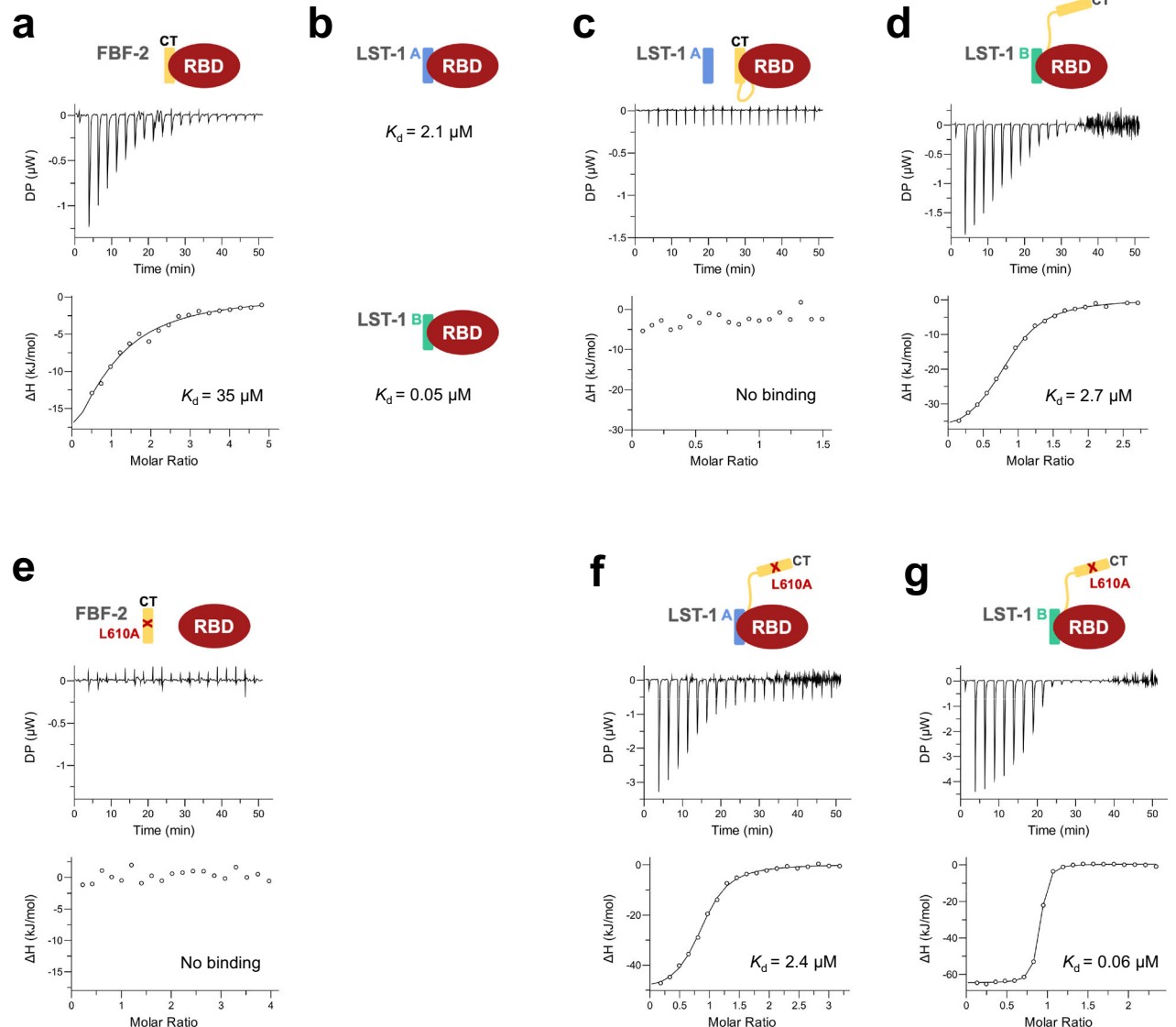

**Fig. 2 | The FBF-2 CT competes with partner proteins for binding to the RBD.**
Representative isothermal titration calorimetry thermograms (top, differential
power [DP] vs time) and corresponding titration curve-fitting graphs (bottom) are
shown for interaction between **a** FBF-2 RBD and FBF-2 CT, **c** FBF-2 RBD + CT and
LST-1 A, **d** FBF-2 RBD + CT and LST-1 B, **e** FBF-2 RBD and FBF-2 CT L610A, **f** FBF-2
RBD + CT L610A and LST-1 A, and **g** FBF-2 RBD + CT L610A and LST-1 B. Data points

on the curve-fitting graphs are shown as open circles. **b** Previously reported mean
$K_d$ from two independent replicates with similar results for interaction between
FBF-2 RBD and LST-1 A (top) and FBF-2 RBD and LST-1 B (bottom)[18]. Mean $K_d$ from
two independent replicates with similar results are indicated. Thermodynamic
parameters are summarized in Supplementary Table 3. Source data are provided as
a Source Data file.

mutant proteins bound to *gld-1* FBEa with similar affinity as RBD alone
($K_d$ = 62 nM for Y479A and 51 nM for L610A vs 70 nM for the RBD alone,
Fig. 3c, d; Supplementary Table 4). Thus, the intramolecular interac-
tion between the CT and RBD of the wild-type protein is required to
lower FBF-2 binding affinity to RNA. We conclude that tethering of the
FBF-2 CT to the RBD is autoinhibitory for RNA binding.

**Electrostatic repulsion by acidic residues in the FBF-2 CT
reduces RNA-binding affinity**
We previously demonstrated that residues upstream of the core PIM in
LST-1 B reduced FBF-2 RBD RNA-binding affinity, and therefore were
interested to know if the region just N-terminal to the intramolecular
PIM of the FBF-2 CT might also inhibit FBF-2 RNA-binding affinity[18]. The
FBF-2 CT IDR is flexible and therefore residues other than the PIM were
not visualized in our crystal structure of FBF-2 RBD + CT bound to RNA
(Fig. 1c). To test the idea that the CT might interfere with RNA binding,
we examined an AlphaFold model of FBF-2 (Fig. 4a)[42,43]. The FBF-2 CT

IDR lacks defined secondary structure in that model, as expected.
Surprisingly, residues LEPR[610–613] within the CT PIM are predicted (with
a high AlphaFold confidence score, predicted Local Distance Differ-
ence Test [pLDDT], of 70 <pLDDT <90) to adopt a conformation
similar to that seen in our crystal structure. The disordered region
between the RBD and PIM forms a loop, which is constrained at the
PIM, but the loop conformations are heterogeneous with low con-
fidence scores of pLDDT<50. This loop places an electronegative
cluster N-terminal to the FBF-2 CT PIM near the 5' end of the bound
RNA (Fig. 4a, b), which might electrostatically interfere with binding to
the negatively charged RNA.

To guide our understanding of the dynamic motions of the FBF-2
CT and 5' RNA, we performed 120 ns molecular dynamics simulations
for the apo and RNA-bound FBF-2 RBD + CT using an AlphaFold-based
starting model docked to an RNA with four additional 5' nucleotides
(Fig. 4a, Supplementary Fig. 2). Although we hypothesized repulsion
between the CT and 5' RNA, we first examined whether attractive

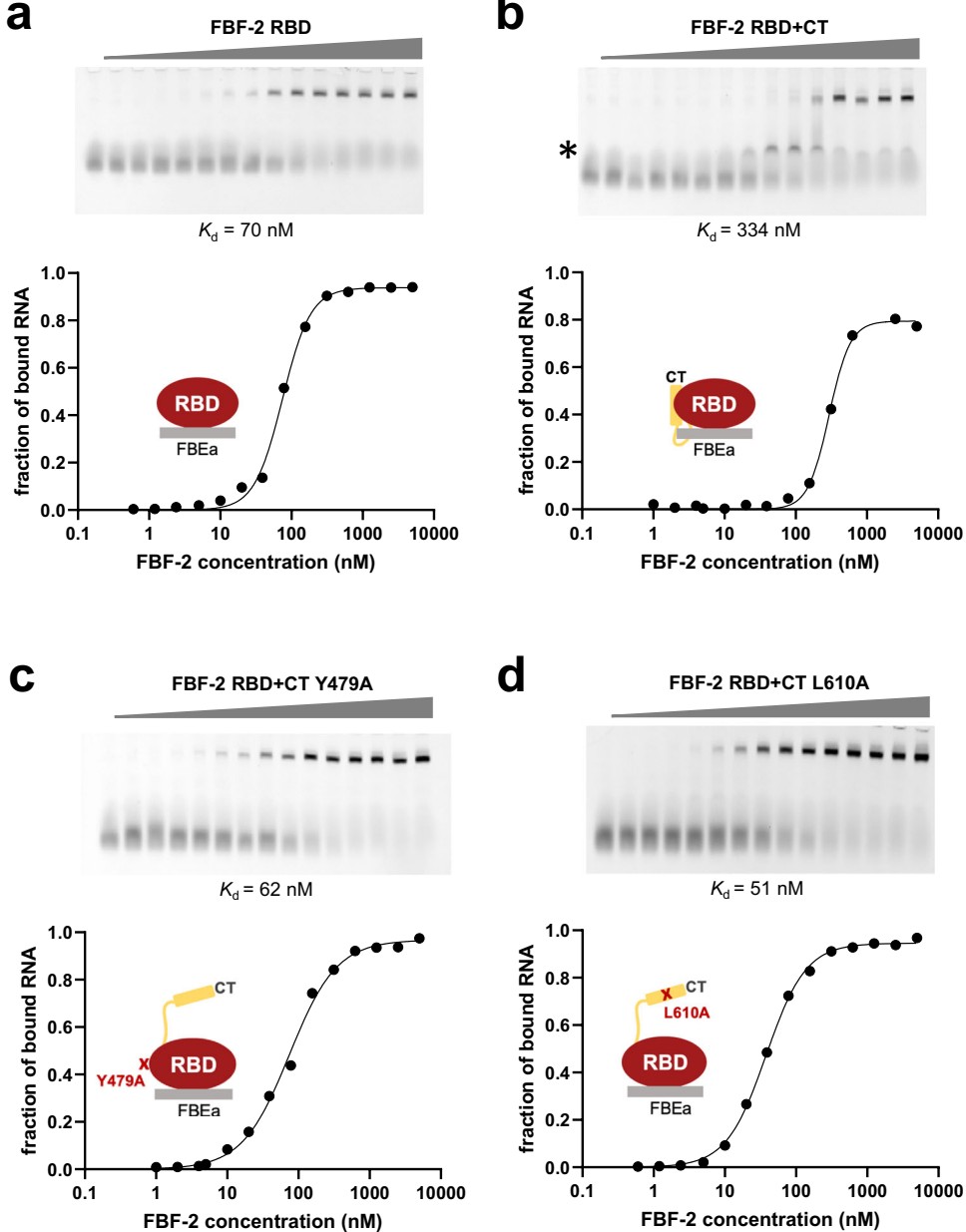

**Fig. 3 | The FBF-2 CT autoinhibits RNA-binding affinity.** Representative EMSA gels and binding curves are shown for binding to FBEa RNA (5′-AUCAU·GUGCCAUAC-3′) by **a** FBF-2 RBD, **b** FBF-2 RBD + CT, **c** FBF-2 RBD + CT Y479A, and **d** FBF-2 RBD + CT L610A. The left lanes are RNA only. Data points on graphs are shown as filled circles. We observed an intermediate band (*) that appears to be a non-specific interaction of FBF-2 RBD + CT that is not observed for the RBD or mutant proteins. We included this band as part of the unbound RNA. Mean $K_d$ from at least three independent technical replicates with similar results are indicated (Number of replicates: FBF-2 RBD, $n = 4$; FBF-2 RBD + CT, $n = 4$; FBF-2 RBD + CT Y479A, $n = 3$; and FBF-2 RBD + CT L610A, $n = 5$). See also Supplementary Table 4. Source data are provided as a Source Data file.

interactions, such as salt bridges, van der Waals interactions, and H-bonds, exist. We observed no salt bridges or van der Waals interactions between the FBF-2 CT IDR and the 5′ end of the bound RNA. We did observe H-bonds for small fractions of time across the trajectories at equilibrium. For example, H593 is the only FBF-2 loop residue that formed an H-bond with the 5′ RNA, and it did so in only 36% of snapshots. Other H-bonds between the FBF-2 CT IDR and the 5′ RNA residues formed for much lower fractions of time (5–10%, Supplementary Table 5). Unlike salt bridges and van der Waals interactions (which we did not observe here), these H-bonds do not contribute to the stability of the complex, because they are readily exchanged by H-bonds with water molecules. We conclude that attractive interactions between the FBF-2 CT IDR and 5′ RNA play little to no role in binding affinity.

We next analyzed the conformational fluctuation of the FBF-2 CT IDR in the presence and absence of bound RNA (Fig. 4c). Since no salt bridges were observed but the two flexible regions were in close proximity, we expected that the bound RNA would reduce flexibility of the FBF-2 CT IDR. Compared with the apo form, RNA-bound FBF-2 has slightly lower conformational fluctuation for the structured RBD (residues 168-575). However, the conformational fluctuation of the FBF-2 CT IDR shows dramatic reduction when bound to RNA (Fig. 4c). This analysis suggests, as predicted, that the 5′ RNA creates a spatial hindrance and reduces the conformational flexibility of the CT IDR. To focus on the global motions in the CT IDR rather than local fluctuations, we performed principal component analysis (PCA). The PCA analysis identified correlated motion between the FBF-2 CT IDR and

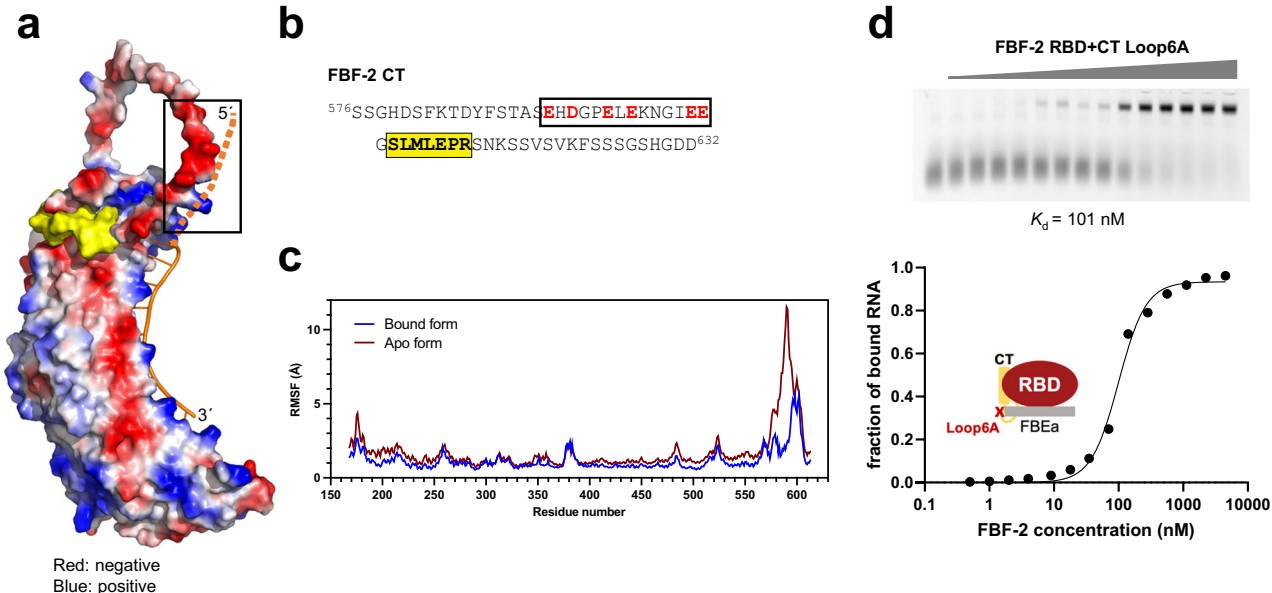

**Fig. 4 | Electrostatic repulsion by acidic residues in the FBF-2 CT reduces RNA-binding affinity. a** Surface representation of an AlphaFold model of FBF-2 RBD + CT colored by electrostatic potential, and the CT PIM is colored yellow. A cartoon of cFBE RNA was added by superimposing the AlphaFold model with the crystal structure of FBF-2 RBD + CT. The path of RNA 5′ to the cFBE is indicated by a dashed line. **b** FBF-2 CT sequence with acidic residues (red) and PIM (yellow). **c** Analysis of conformational fluctuations in FBF-2 RBD + CT. Root mean square fluctuations (RMSF) are plotted by FBF-2 RBD + CT residues for apo and RNA bound states. **d** Representative EMSA gel and binding curve are shown for binding of FBF-2 RBD + CT Loop6A mutant to FBEa RNA. The left lane is RNA only. Data points on the graph are shown as filled circles. Mean $K_d$ from three independent technical replicates with similar results is indicated. See also Supplementary Table 4. Source data are provided as a Source Data file.

the 5′ RNA (Supplementary Movie 1). Considering the absence of consistent attractive interactions between these two regions, we interpret the correlated motions as arising from 'pushing' or electrostatic repulsion.

To test the importance of the cluster of charged residues and predicted repulsion, we substituted the relevant acidic residues in the CT (Fig. 4b) with alanine, and we refer to this mutated protein as RBD + CT Loop6A to refer to the six alanine mutations in the loop. These substitutions only modestly decreased the thermal stability of the protein (Supplementary Table 1), since the FBF-2 CT PIM in this mutant is intact and can still bind to the RBD. The acidic residues are not important for competition with LST-1 binding, as the Loop6A mutant reduced the affinity of interaction of FBF-2 RBD + CT with LST-1 B, similar to that of wild type protein (Supplementary Table 3). We measured the RNA-binding affinity of RBD + CT Loop6A to *gld-1* FBEa RNA. The reduced negative charge in the FBF-2 CT loop increased binding affinity to RNA ($K_d$ = 334 nM for RBD + CT vs 101 nM for RBD + CT Loop6A, Fig. 4d) and thus diminished CT autoinhibition. Therefore, both tethering of the CT and the cluster of charged residues contribute to CT autoinhibition.

**FBF-2 CT accelerates dissociation from the RNA**
To further examine the effect of the CT on RNA-binding activity of FBF-2, we performed surface plasmon resonance (SPR) assays to measure the kinetics of FBF-2 binding to *gld-1* FBEa RNA. We first compared the binding kinetics of FBF-2 RBD and FBF-2 RBD + CT. We found that the on rate for FBF-2 RBD binding was slightly faster than for RBD + CT: $13.6 \times 10^4\,M^{-1}\,s^{-1}$ vs $7.1 \times 10^4\,M^{-1}\,s^{-1}$. In contrast, the off rate for RBD + CT was -5-fold faster than for RBD: $4.42 \times 10^{-3}\,s^{-1}$ vs $0.86 \times 10^{-3}\,s^{-1}$ (Fig. 5), indicating that the CT accelerates dissociation from the *gld-1* FBEa RNA. We therefore measured the binding kinetics of FBF-2 RBD + CT mutants, Y479A, L610A, and Loop6A. Consistent with the differences in off rate for FBF-2 RBD vs RBD + CT and our EMSA results above, the off rates for the RBD + CT mutants were all slower than for wild type

RBD + CT and similar to the off rate of FBF-2 RBD alone (Fig. 5c, Supplementary Fig. 3). As noted above, the electrostatic repulsion by acidic residues in the CT is diminished when the CT loop is not constrained near the RNA (Y479A and L610A) or the cluster of charged residues are substituted with alanine (Loop6A). The SPR data further support the conclusion that the FBF-2 CT lowers RNA-binding affinity and does so by destabilizing protein-RNA interaction and accelerating dissociation.

**LST-1 reverses FBF-2 CT autoinhibition of RNA-binding affinity**
If the FBF-2 CT reduces RNA-binding affinity (Fig. 3b) and competes with LST-1 for binding (Fig. 2c, d), why then does FBF-2 depend on LST-1 to repress RNAs in nematodes? To address this question, we explored RNA-binding affinities of FBF-2 RBD + CT in the presence of either LST-1 A or LST-1 B (Fig. 6; Supplementary Table 3). In contrast to results with FBF-2 RBD alone[17,18], both LST-1 A and LST-1 B increased the RNA-binding affinity of FBF-2 RBD + CT. LST-1 A increased the RNA-binding affinity of FBF-2 RBD + CT to FBEa RNA 3-fold ($K_d$ = 114 nM, Fig. 6a), although it had no effect on the RNA-binding affinity of RBD[18]. Similarly, LST-1 B increased the RNA-binding affinity of FBF-2 RBD + CT to FBEa RNA 5-fold ($K_d$ = 68 nM, Fig. 6b), although it decreased the RNA-binding affinity of FBF-2 RBD. Remarkably, the RNA-binding affinity of FBF-2 RBD + CT together with either LST-1 A or B was similar to that of the RBD alone (70 nM, Fig. 3a). As expected, LST-1 B has no effect on the RNA-binding affinity of FBF-2 RBD + CT Y479A (Supplementary Fig. 1b), as the mutation disrupts LST-1 binding[17]. We also found that LST-1 B had little to no effect on the RNA-binding affinity of FBF-2 RBD + CT L610A (Supplementary Fig. 1c) and FBF-2 RBD + CT Loop6A (Supplementary Fig. 1d), although protein-protein interactions were not affected (Supplementary Table 3). Together, our results suggest that LST-1 A and B can displace the FBF-2 CT and thereby reverse its intramolecular autoinhibitory effect on RNA-binding affinity.

## Discussion

Here we identify a molecular mechanism for the FBF-2 and LST-1 interdependence in *gld-1* repression in germline stem cells and shed light on how IDRs regulate RNA-binding affinity. We demonstrate that the FBF-2 high-affinity, sequence-specific binding to target RNAs is modulated in two ways: (1) FBF-2 in vitro RNA-binding affinity is autoinhibited by its intrinsically disordered CT, and (2) this auto-inhibition is relieved by interaction with either PIM in the partner protein LST-1. We further show that autoinhibition by the CT is imposed by interaction of the RBD with an intramolecular PIM in the CT. Yet how does this intramolecular interaction result in reduced RNA-binding affinity? We propose that the FBF-2 PIM acts as Velcro®, attaching to the RBD and constraining the disordered loop to be located near the 5′ end of the bound RNA where an electronegative cluster of acidic residues in the CT weakens RNA-binding affinity by electrostatic repulsion.

FBF-2 requires LST-1 to maintain GSCs, and this partnership depends on LST-1's two PIMs, LST-1 A and LST-1 B. We suggest that LST-1 interaction relieves FBF-2 autoinhibition resulting in increased RNA-binding affinity. LST-1 A and LST-1 B bind with higher affinity to FBF-2 than its CT PIM and will outcompete the FBF-2 CT PIM. When the FBF-2 CT PIM is released by LST-1 competition, we propose that the FBF-2 CT loop is no longer formed and therefore the CT with its electronegative cluster does not interfere with RNA binding.

Although many RNA-binding proteins contain IDRs that influence RNA binding, FBF-2 autoinhibition by its CT and relief by LST-1 is distinct from other reported control mechanisms. IDRs, such as serine/arginine-rich regions in SR proteins and arginine-glycine repeats in FMRP, have been shown to directly interact with RNA[44,45], and others, such as the C-terminal tail of Drosophila stem-loop-binding protein (SLBP) and an N-terminal extension to the YTH domain of fission yeast Mmi1, promote or stabilize RNA-bound protein conformations to increase RNA-binding affinity[46,47]. IDRs in RNA-binding proteins can also inhibit RNA binding, typically via electrostatic interactions mediated by acidic residues. For example, the C-terminal acidic tail of the *E. coli* RNA chaperone protein Hfq inhibits non-specific RNA binding by interacting with basic patches on the protein[48], and a poly D/E sequence in yeast Nop15 stabilizes the neighboring RRM and suppresses non-specific RNA binding[49]. FBF-2 also utilizes an IDR with an electronegative cluster to interfere with RNA binding. In this case, however, inhibition controls its specific RNA-binding activity, and LST-1 interaction reverses autoinhibition and increases RNA-binding affinity.

Moreover, FBF-2 and its paralog FBF-1 function throughout the germline along with additional PUF partner proteins, such as CPB-1 and GLD-3[50,51]. Both CPB-1 and GLD-3 harbor consensus KxxL motifs suggesting they interact with FBF-2 via the same surface as LST-1 does[52–54]. Therefore, CPB-1 and GLD-3 likely collaborate with FBF-2 similar to LST-1: they could also displace the FBF-2 CT and modulate the RNA-binding activity of FBF-2. Different interaction strengths of the FBF-2 CT vs partner proteins with the FBF-2 RBD appear to be key to regulating the RNA-binding activity of FBF-2. The equilibrium of competitive binding can be shifted by altering the local abundance of partner proteins, although the effective concentration of the intramolecular CT is constant. Thus, the RNA-binding affinity of FBF-2 can be finely tuned in different regions of the *C. elegans* germline.

*C. elegans* express nine PUF proteins, including FBF-2. Are the other eight PUF proteins regulated by partner proteins with PIMs and are any of them regulated via an intramolecular PIM? A *C. elegans* 'PUF hub' identifies additional protein partnerships mediated by PIMs that are essential for germline stem cell self-renewal[55]. This hub includes four PUF proteins (FBF-1, FBF-2, PUF-3, and PUF-11) and two partner proteins (LST-1 and SYGL-1). LST-1 and SYGL-1 interact with the four PUF proteins[21,55]. Two PIMs have been identified in both LST-1 and SYGL-1, and they are essential for germline stem cell maintenance[22,23,55],

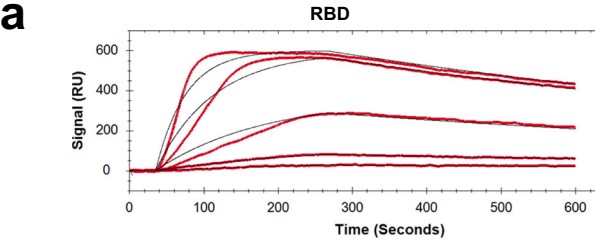

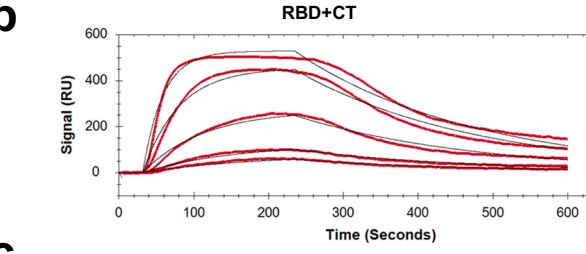

| FBF-2 | $k_{on}$ (x10$^4$ M$^{-1}$ s$^{-1}$) | $k_{off}$ (x10$^{-3}$ s$^{-1}$) | $K_d$ (nM) |
|---|---|---|---|
| RBD | 13.6 ± 2.2 | 0.86 ± 0.16 | 6.6 ± 2.3 |
| RBD+CT | 7.1 ± 1.7 | 4.42 ± 0.39 | 65.1 ± 21.1 |
| RBD+CT Y479A | 10.2 ± 4.3 | 1.18 ± 0.11 | 12.9 ± 6.4 |
| RBD+CT L610A | 15.5 ± 4.7 | 1.36 ± 0.11 | 9.4 ± 3.6 |
| RBD+CT Loop6A | 10.9 ± 4.3 | 0.94 ± 0.05 | 9.5 ± 4.2 |

**Fig. 5 | Kinetics of the interaction between FBF-2 and FBEa RNA.**
**a** Representative SPR binding curves for FBF-2 RBD. Five protein concentrations were run, shown from top to bottom: 200, 100, 50, 25, and 12.5 nM (red lines), and the curves were simultaneously fitted to a 1:1 binding model (black lines).
**b** Representative SPR binding curves (red) and fitted curves (black) for FBF-2 RBD + CT. Five protein concentrations were run, shown from top to bottom: 400, 200, 100, 50, and 25 nM. **c** Table of on/off rates and derived $K_d$ values for RBD and RBD + CT variants. Values are mean ± SD from two technical replicates with similar results. Source data are provided as a Source Data file.

yet the molecular interactions and impact on RNA-binding affinity have been studied only for the FBF-2 and LST-1 partnership[17,18]. Below we summarize what is known about *C. elegans* PUF proteins and their potential to be regulated by inter- or intramolecular PIMs (Supplementary Fig. 4).

In the PUF hub, FBF-1 and FBF-2 play a more prominent role than PUF-3 and PUF-11[55]. FBF-1 and FBF-2 amino acid sequences are 91% identical and have largely overlapping functions. However, genetic experiments suggest that FBF-1 and FBF-2 also have distinct functions[41,56,57] and that the FBF-2 CT is associated with some FBF-2-specific functions compared to FBF-1[57]. For example, when expressed as the sole FBF protein, FBF-2 produces shorter germline progenitor regions than FBF-1. When the CT of FBF-1 is replaced with the FBF-2 CT, the chimeric protein produces a progenitor region that matches the shorter length produced by FBF-2. The FBF-2 CT (residues 570-632), including its PIM, is longer than the FBF-1 CT (residues 568–614). The FBF-1 CT includes an electronegative cluster whose sequence is almost identical to FBF-2. This highly conserved sequence is followed by a sequence that might be a PIM, [605]NLRLMRT[611], but it is quite different from the FBF-2 PIM (Supplementary Fig. 4a). It therefore must be tested experimentally whether this FBF-1 sequence is a PIM and whether it has a regulatory function.

The sequence in FBF-1 raises an important point: PIMs are difficult to predict from amino acid sequences, with a leucine residue one of the limited sequence features. Recently, SYGL-1 was shown to have two

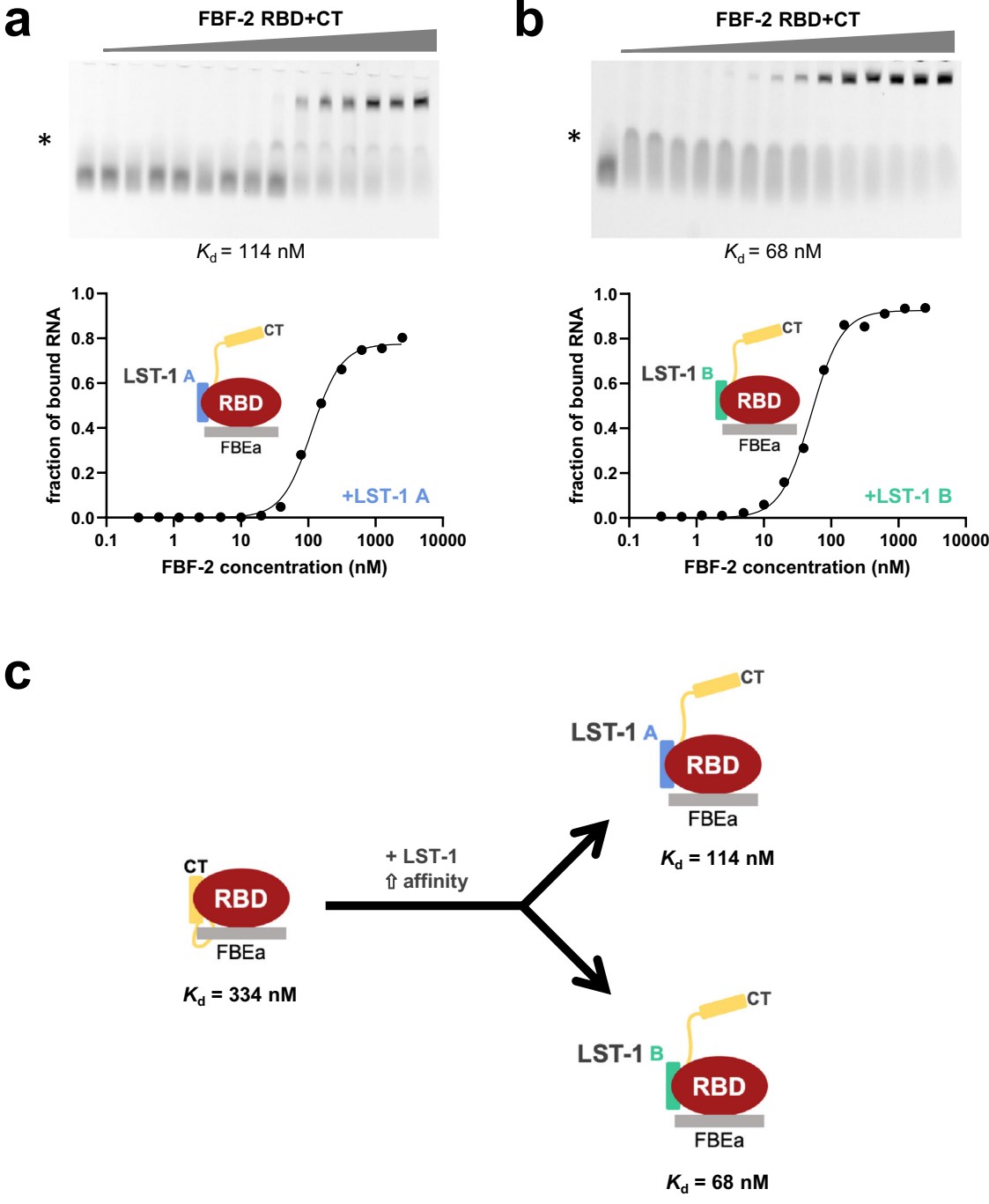

**Fig. 6 | LST-1 A and B release autoinhibition and increase RNA-binding affinity of FBF-2 RBD + CT.** Representative EMSA gels and binding curves are shown for binding to FBEa RNA by **a** FBF-2 RBD + CT in the presence of 150 μM LST-1 A and **b** FBF-2 RBD + CT in the presence of 50 μM LST-1 B. The concentrations of LST-1 A and LST-1 B were chosen based on the affinities for FBF-2; LST-1 A binds with weaker affinity than LST-1 B. The left lanes are RNA only. Data points on the graphs are shown as filled circles. We observed an intermediate band (*) that appears to be non-specific interaction of FBF-2 RBD + CT **a** or LST-1 B **b**, as it occurs at low protein concentration. We included this band as part of the unbound RNA. Mean $K_d$ from three independent technical replicates with similar results are indicated. See also Supplementary Table 4. Source data are provided as a Source Data file. **c** Cartoon illustrating relief of FBF-2 CT autoinhibition by LST-1 A or LST-1 B.

non-identical PIMs that are essential for GSC maintenance[23]. As we found for the FBF-2 CT PIM, PIMs need not possess a strict KxxL motif: SYGL-1 A contains the sequence KQIL$_{42}$TLK, but SYGL-1 B contains VTLL$_{80}$ELK. The critical leucine residues (L42 and L80) were identified by testing in vivo function, but SYGL-1 B has a valine rather than a lysine in the first position. It appears that PIM leucine residues must be in a sequence or structural context, but we do not yet understand it. For example, SYGL-1 PIMs were not functional when the critical leucines were mutated to alanine, despite the presence of neighboring leucine residues[23]. We also observed here that FBF-2 L608 (Supplementary Fig. 4a) did not substitute for L610 to autoinhibit FBF-2 RNA-binding affinity when we mutated L610 to alanine (Fig. 3e).

The longer FBF-2 CT also includes a serine-rich region following the PIM (Supplementary Fig. 4a), absent from FBF-1, which is predicted to be phosphorylated[58,59]. Phosphorylation of the FBF-2 CT offers additional opportunities to regulate FBF-2 vs FBF-1. Phosphorylation would alter the charge of the FBF-2 CT, and this reversible electronegative cluster may reduce RNA-binding affinity by repelling

negatively charged RNA or perhaps modify RBD and PIM interaction. The longer FBF-2 CT could also recruit additional protein factors that would be different for FBF-1. Genetic experiments suggest that FBF-1 function is dependent on the CCR4-NOT deadenylase complex but that the FBF-2 CT confers independence from CCR4-NOT[57]. In addition, FBF-2 appears to interact with dynein light chain DLC-1, but FBF-1 does not[41]. Perhaps protein factors recruited by the FBF-2 CT produce this independence.

PUF-3 and PUF-11 play minor roles in the PUF hub[55], seeming to supplement the activities of FBF-1 and -2. They are more similar to each other (89% identical residues) than to FBF-1 and FBF-2 (39% identical residues). Nevertheless, their RNA sequence specificities are similar to FBF-1 and FBF-2. An in vitro selection experiment identified three classes of RNA sequences bound by PUF-11[60], and we now recognize that two of these classes are comparable to the 9-nt *gld-1* FBEa and the 8-nt cFBE[17,36]. This overlapping sequence specificity suggests that the four hub PUF proteins can regulate similar target mRNAs. However, the mechanistic details of the partnership between PUF-3/-11 and LST-1/SYGL-1 appear to be different than for FBF-2. Although the residues that are important for the recognition of LST-1 by FBF-2 are also found in PUF-3 and PUF-11 (Supplementary Fig. 4b, c), PUF-3 and PUF-11 have no residues following the RBD and therefore no autoinhibitory C-terminal PIM (Supplementary Fig. 4a). Without the autoinhibition, LST-1/SYGL-1 interaction with PUF-3/PUF-11 may modulate RNA-binding affinity directly by interacting with the RBD near the 5' end of the RNA.

In addition to the PUF proteins in the GSC self-renewal PUF hub, *C. elegans* express two additional classes of phylogenetically related PUF proteins: PUF-8, -9 and PUF-5, -6, -7. PUF-8, and PUF-9 are like Drosophila Pumilio and mammalian PUM1 and PUM2. They recognize the 8-nt sequence, 5'-UGUAnAUA-3', where 'n' is any nucleotide, and they lack a pocket to recognize an upstream cytosine[40,61]. Residues in FBF-2 that are important for interaction with LST-1 are largely absent from PUF-8 and PUF-9 (Supplementary Fig. 4b, c). Therefore, it seems unlikely that PUF-8 and PUF-9 interact with PIMs, unless their corresponding PIMs have different features. Nevertheless, PUF-9 ends with a serine-rich region like FBF-2 that could also be phosphorylated, but its equivalent is absent from PUF-8 (Supplementary Fig. 4a).

PUF-5, -6, and -7 are likely to interact with PIMs in partner proteins like LST-1 and SYGL-1, as interacting residues equivalent to those in FBF-2 are present (Supplementary Fig. 4b, c). PUF-6 and PUF-7 are highly similar (98% identical residues), and PUF-5 is closely related but more divergent (54% identical residues). PUF-5 and PUF-6 (and by extension, PUF-7) recognize the identical high affinity consensus sequence, 5'-CUCUGUAUCUUGU-3', that contains two UGU elements and an upstream cytosine (5' to the first UGU)[40,62]. CT PIMs in PUF-5, -6, and -7 with a KxxL motif are not present, but given the difficulty in predicting PIMs, we note that leucine residues, including a sequence in PUF-6 and PUF-7 that resembles the FBF-2 PIM, are present that must be tested for function (Supplementary Fig. 4a).

The *C. elegans* PUF proteins include four subclasses of RNA-binding proteins with distinct RNA sequence specificity, and distributed among them are different sets of potential regulatory elements: PIM binding residues, CT autoregulatory PIM, acidic cluster, or serine cluster. We now understand the interdependence of FBF-2 and LST-1 through PIM interaction and autoinhibition, yet the other hub PUF proteins bind to LST-1 and SYGL-1, but may not be autoinhibited. Therefore, additional modes of partnerships between PUF proteins and LST-1/SYGL-1 must exist. *C. elegans* PUF proteins also include PUF-8, -9 that do not appear to be regulated by PIMs, and PUF-5, -6, -7 that do. It appears we have only begun to uncover the ways *C. elegans* PUF and partner proteins combine different mechanisms to control target RNAs.

## Methods
### Protein expression and purification

A cDNA fragment encoding *C. elegans* FBF-2 RBD + CT (residues 164-632) was cloned into the pSMT3 vector (kindly provided by Dr. Christopher Lima), which encodes an N-terminal His$_6$-SUMO tag followed by a Ulp1 protease cleavage site[63]. The FBF-2 RBD + CT protein was overexpressed in *E. coli* BL21-CodonPlus(DE3)-RIL competent cells (Agilent). A 1-L LB culture with 50 μg/mL kanamycin was inoculated with a 5-mL overnight culture and grown at 37 °C to OD$_{600}$ of ~0.6. Protein expression was induced with 0.1 mM IPTG, and the culture was grown at 16 °C overnight. The cell pellet was resuspended in 40 ml lysis buffer containing 20 mM Tris, pH 8.0; 0.5 M NaCl; 20 mM imidazole; 5% (v/v) glycerol; and 0.1% (v/v) β-mercaptoethanol and disrupted by sonication. After centrifugation, the His$_6$-SUMO-tagged FBF-2 RBD + CT protein was purified from the soluble fraction of the *E. coli* cell lysate in lysis buffer using Ni-NTA resin (Qiagen), and was eluted with a buffer containing 20 mM Tris pH 8.0, 50 mM NaCl, 0.2 M imidazole and 1 mM DTT. The eluted fusion protein was incubated with Ulp1 protease overnight to remove the His$_6$-SUMO tag. Subsequently, the FBF-2 protein was purified with a Hi-Trap Heparin column (Cytiva), eluting with a 5–100% gradient of buffer B. Heparin column buffer A contained 20 mM Tris pH 8 and 1 mM DTT, and buffer B contained an additional 1 M NaCl. The peak fractions from the heparin column were pooled and concentrated to 5 ml and loaded onto a HiLoad 16/60 Superdex 75 column (Cytiva) in a buffer containing 20 mM HEPES pH 7.4, 150 mM NaCl, and 0.5 mM TCEP. Purified FBF-2 RBD + CT protein was concentrated and snap-frozen for later binding experiments.

Mutants of FBF-2 RBD + CT: Y479A (5'-GACGAGATTTTCG ACGGAGCCATTCCACATCCGGACAC), L610A (5'-GGAAGGAAGCCTGA TGGCAGAGCCACGGAGCAAT), and Loop6A (E592A, D594A, E597A, E599A, E604A, E605A; 5'-CAGTACTGCTTCCGCGCACGCTGGTCCG GCGTTGGCGAAGAATGGGATCGCGGCAGGAAGCCTGATGC) were generated by site-directed mutagenesis with the primers indicated. Plasmid sequences were verified by DNA sequencing (Azenta Life Sciences). Protein variants were expressed and purified like wild type proteins. Melting temperatures of wild type and mutant proteins were assessed by differential scanning calorimetry. Reaction mixtures (total volume of 20 μl) contained ~0.3 mg/ml FBF-2 proteins and Sypro Orange dye (1:1000 dilution) (Sigma). Fluorescent intensity was collected from 25 °C to 95 °C (3 °C increment/min) with a real time PCR instrument (Applied Biosystems™ QuantStudio 7 Flex System) using excitation and emission wavelengths of 470 nm and 586 nm, respectively. Protein Thermal Shift™ software (Applied Biosystems, version 1.4) was used to analyze protein melting curves and calculate melting temperatures (T$_m$).

The FBF-2 RBD protein (residue 164-575) was purified as described previously[61], using a plasmid with a cDNA fragment encoding the FBF-2 RBD cloned into the pSMT3 vector (kindly provided by Dr. Christopher Lima). This plasmid encodes an N-terminal His-SUMO fusion tag followed by a TEV protease cleavage site[63]. TEV protease was used to cleave the His-SUMO tag, instead of Ulp1 protease, because it was generated from a previous pGEX6p FBF-2 RBD construct. Otherwise, the purification procedure was the same as the FBF-2 RBD + CT protein described above.

LST-1 A and B peptides were purified as described previously[18], using plasmids with cDNA fragments encoding LST-1 A (residues 19–50) and LST-1 B (residues 67–98) cloned into the pSMT3 vector. A 1 L culture of TB media with 50 μg/mL kanamycin was inoculated with a 5-mL culture grown overnight at 37 °C. The 1-L culture was grown at 37 °C. At OD$_{600}$ of ~1.0, protein expression was induced with 0.4 mM IPTG, and the culture was grown at 22 °C for ~20 h. After lysis by sonication and centrifugation, the soluble fraction of *E. coli* cell lysate in a buffer containing 20 mM Tris, pH 8.0; 0.5 M NaCl; 20 mM imidazole; and 5% (v/v) glycerol was mixed with 5 mL Ni-NTA resin (Qiagen) for 1 h at 4 °C. After extensive washing the LST-1 proteins were eluted

with a buffer of 20 mM Tris, 50 mM NaCl, and 200 mM imidazole, pH 8. Ulp1 protease was added to the eluant and incubated at 4 °C for 2 h or overnight to remove the His$_6$-SUMO tag from LST-1 A or B. LST-1 A protein was separated from His$_6$-SUMO with a HiTrap Q column (Cytiva), and the column flow-through, which contained LST-1 A, was collected and concentrated using Amicon Ultra-15 filters (3 K MWCO). LST-1 B protein was purified with a HiTrap Heparin column (Cytiva) and eluted with a 5–100% NaCl gradient (buffer A: 20 mM Tris, pH 8.0; buffer B: 20 mM Tris, pH 8.0, 1 M NaCl). The peak fractions containing LST-1 B were pooled and concentrated. Both LST-1 A and B were further purified with a HiLoad 16/60 Superdex 75 column (Cytiva).

LST-1 protein concentrations were determined by NanoDrop based on UV absorption at 280 nm.

## Purification of FBF-2 CT peptide

A cDNA fragment encoding FBF-2 CT (residues 601–632) was cloned into the pSMT3 vector. The protein was overexpressed in *E. coli* BL21-CodonPlus (DE3)-RIL competent cells (Agilent). A 1-L TB culture was inoculated with a 5-mL overnight culture and grown at 37 °C to OD$_{600}$ of -1.0 and then grown at 22 °C overnight after induction with 0.4 mM IPTG. The soluble fraction of *E. coli* cell lysate in a buffer containing 20 mM Tris, pH 8.0; 0.5 M NaCl; 20 mM imidazole; and 5% (v/v) glycerol was mixed with 5 mL Ni-NTA resin for 1 h at 4 °C. After extensive washing the FBF-2 CT proteins were eluted with a buffer of 20 mM Tris, 50 mM NaCl and 200 mM imidazole, pH 8. The Ulp1 protease was added to the eluant and incubated at 4 °C for 1 h to remove the His$_6$-SUMO tag. FBF-2 CT protein was purified with a HiTrap Q column, and the column flow-through containing FBF-2 CT was collected and concentrated using Amicon Ultra-15 filters (3 kDa MW cutoff). It was further purified with a HiLoad 16/60 Superdex 75 column. The FBF-2 CT protein concentration was determined using the Qubit protein assay kit (Invitrogen) because it lacks aromatic residues.

## Crystallization, data collection, and structure determination

Purified FBF-2 RBD + CT protein (OD$_{280}$ = 3.62) was incubated with cFBE RNA (5′-CUGUGAAUG-3′) at a molar ratio of 1:1.2 on ice for 1 h prior to crystallization. Crystals of the protein–RNA complex were grown in 30% PEG 400, 0.1 M Tris pH 8.5 by hanging drop vapor diffusion at 20 °C with a 1:1 ratio of sample:reservoir solution. Crystals were directly flash frozen in liquid nitrogen.

X-ray diffraction data were collected at a wavelength of 1.0 Å using SERGUI at beamline 22-ID (SER-CAT) of the Advanced Photon Source. Data sets were scaled with HKL2000[64]. Crystals belonged to the P6$_1$ space group. An asymmetric unit contained one ternary complex. The structure of the FBF-2 RBD/*gld-1* FBEa binary complex (PDB code: 3v74) was used as a search model for molecular replacement with Phaser[65] as implemented in Phenix version 1.20–4459. For the FBF-2 RBD + CT/cFBE complex, the C-terminal peptide of FBF-2 (residues 607–613) was built manually into the electron density. Residues 570–606 were disordered. The model was improved through iterative refinement and manual building with Phenix version 1.20–4459 and Coot (version 0.9.6 EL)[66,67]. Data collection and refinement statistics are shown in Supplementary Table 2.

## Isothermal titration calorimetry

Experiments were performed at 20 °C using a MicroCal PEAQ-ITC Automated (Malvern Instruments) with a 200-µL standard cell and a 40-µL titration syringe. FBF-2 and LST-1 variants were prepared in a buffer of 20 mM HEPES pH 7.4, 150 mM NaCl, and 0.5 mM TCEP by gel filtration. LST-1 (100–400 µM) was titrated from the syringe into the cell containing FBF-2 (10–30 µM) in 2 µl aliquots. Experiments were performed in duplicate due to the limitation of the amount of protein needed. Data were analyzed using the MicroCal PEAQ-ITC Analysis Software (version 1.40) provided by the manufacturer with the one-site model (Table S3).

## Electrophoretic mobility shift assays

A 3′-Cy5-labeled synthetic RNA was ordered from IDT (*gld-1* FBEa, 5′-AUCAUGUGCCAUAC-3′). The FBF-2 proteins were serially diluted 2-fold from the highest concentration of 10 µM in the buffer used for gel filtration (20 mM HEPES pH 7.4, 150 mM NaCl, and 0.5 mM TCEP). 10 µl of the serially diluted protein samples were mixed with 10 µl RNA (10 nM) in a 2× buffer of 10 mM HEPES pH 7.4, 0.01% (v/v) Tween 20, 0.1 mg/ml BSA, 0.1 mg/ml yeast tRNA, and 2 mM DTT. For experiments in the presence of LST-1, LST-1 proteins at constant concentration were preincubated with FBF-2 at 4 °C for 2 h. The LST-1 concentrations were chosen based on their $K_d$'s for interaction with FBF-2. The final FBF-2 concentrations ranged from 5000 nM to 0.6 nM, and the LST-1 A or B concentrations were constant at 150 µM or 50 µM, respectively. The protein-RNA mixtures were incubated at 4 °C overnight. The samples were resolved on 10% TBE polyacrylamide gels run at constant voltage (100 V) with 1× TBE buffer at room temperature for 35 min. The gels were scanned and visualized with a Typhoon FLA 9500 imager using the Cy5 channel (excitation wavelength 635 nm). Band intensities were quantified with ImageQuant 5.2 (Cytiva). The data were fit with GraphPad Prism (version 9.2.0) using nonlinear regression with a one-site specific binding model. Mean $K_d$'s and standard error of the mean (SEM) from three or more technical replicates are reported (Supplementary Table 4). Full scan images for the replicates shown in the main figures and gel images for all replicates are included in a source data file.

## Molecular dynamics simulations

To build a model containing the FBF-2 RBD + CT bound to RNA, including residues 570–606 that were disordered in our crystal structure, we obtained the AlphaFold structure of FBF-2 protein from UniProt (UniProt Entry: Q09312). We selected residues 168–613 of the AlphaFold structure[42,43], which includes residues 168–569 that are predicted with high confidence and extends through the CT PIM, and aligned the AlphaFold structure with our crystal structure of FBF-2 RBD + CT bound to cFBE RNA in PyMol. The RMSD for residues 168–613 was 0.50 Å over 373 CA atoms. Using this alignment of the AlphaFold and crystal structures, we calculated the RMSD for residues 608–613, which was 1.35 Å over 7 CA atoms. We then removed the protein components of the crystal structure to obtain an initial FBF-2 RBD + CT (residues 168–613)/cFBE RNA complex. Four additional ribonucleotides, AUAU, were added to 5′ end, and this protein-RNA complex was used as the starting model for the molecular dynamics (MD) simulation. An MD simulation for the Apo form of FBF-2 RBD + CT was also run for comparison.

The Amber20 package with AMBER force fields ff19SB, RNA.OL3 and water.opc was used for MD simulations[68]. The charge of the complex or apoprotein was neutralized with corresponding amount of Na$^+$ or Cl$^-$ and then solvated with 150 mM NaCl (Supplementary Table 6). The size of the simulated systems was approximately 14 nm × 13 nm × 13 nm. The MD simulation protocol is as follows for both simulations[69]: (1) steepest descent energy minimization of the solvent water with restraints on the protein and ions; (2) 20 ps constant number-pressure-temperature (NPT) MD simulation at 50 K and 1 atm to equilibrate solvent water with restrains on the protein and ions; (3) heating up the system to 300 K via a series of 10-ps constant number-volume-temperature (NVT) MD simulations at 50, 100, 150, 200, 250, and 300 K; (4) 120-ns production NPT MD simulation at 300 K and 1 atm. In the production MD simulations, a 2-fs time step was used with SHAKE constraints on all bonds involving hydrogen. Long-range electrostatic interactions were treated with the particle-mesh Ewald method. The cutoff for the Lennard-Jones potential was set at 1.0 nm. The simulations were run three times with different starting conformations, and the RMSF analysis is shown in Supplementary Fig. 2. The three runs revealed a consistent flexible region located between residues 580–600. The root mean square deviations (RMSD)

analysis was performed on Cα atoms to determine when the systems are in equilibrium (Supplementary Fig. 2). The equilibrated portion of the trajectories was used to calculate root mean square fluctuations (RMSF) of each residue (Fig. 4c). A total of 12,000 frames were analyzed.

## Surface plasmon resonance assays

Kinetics of FBF-2 binding to the *gld-1* FBEa RNA were measured using a 2-channel OpenSPR instrument (Nicoya). A 3′-biotin-labeled synthetic RNA was ordered from Horizon Discovery (*gld-1* FBEa, 5′-AUCAU-GUGCCAUACA-biotin-3′). The 3′-biotinylated RNA was immobilized on streptavidin-treated biotin sensor chips (Nicoya). All experiments were performed at 20 °C with a buffer of 20 mM HEPES, pH 7.4, 150 mM NaCl, and 2 mM DTT. SPR data were collected using OpenSPR (Nicoya, version 4.3). FBF-2 proteins were injected with a flow rate of 20 μl/min. Data were obtained at five different protein concentrations (25, 50, 100, 200, and 400 nM for wild type RBD + CT; 12.5, 25, 50, 100, and 200 nM for RBD + CT mutants and wild type RBD). Sensor chips were regenerated using 0.05% or 0.1% (v/v) SDS. For each experiment, five binding curves were simultaneously fit using a 1:1 binding model with the TraceDrawer software (Ridgeview Instruments, version 1.9.1). Mean values and standard deviation were calculated from two technical replicates. Despite our efforts, the fit of the binding curves for the association phase of FBF-2 RBD and FBF-2 RBD + CT mutants is poorer than the fit for FBF-2 RBD + CT WT. The appearance of the curves is consistent with the derived $k_{on}$ and $k_{off}$ values, and importantly, the curve fit for the dissociation phase is better.

## Reporting summary

Further information on research design is available in the Nature Portfolio Reporting Summary linked to this article.

## Data availability

The atomic coordinates and structure factors generated in this study for the reported crystal structure have been deposited in the Protein Data Bank under accession number 8SJ7. PDB ID 3V74 was used as a model for molecular replacement. The starting and ending coordinates for the molecular dynamics simulations are available as Supplementary Data 1 and 2, respectively. The molecular dynamics initial and extension input files are available as Supplementary Data 3 and 4, respectively. All other data supporting the findings of this study are available within the paper and its Supplementary Information. Source data are provided in this paper.

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

## Acknowledgements

We thank Lars Pedersen for crystallographic and data collection support at NIEHS. Data were collected at the Southeast Regional Collaborative Access Team (SER-CAT) 22-ID beamline at the Advanced Photon Source, Argonne National Laboratory. SER-CAT is supported by its member institutions, and equipment grants (S10_RR25528, S10_RR028976, and S10_OD027000) from the National Institutes of Health. Use of the Advanced Photon Source was supported by the U.S. Department of Energy, Office of Science, Office of Basic Energy Sciences, under Contract No. W-31-109-Eng-38. We are grateful to Judith Kimble, Sarah Crittenden, Brian Carrick, and Ahlan Ferdous for helpful discussion and comments on the manuscript. We thank our NIEHS colleagues Geoffrey Mueller, Lalith Perera, and Huanchen Wang for the critical reading of the manuscript. This work was supported in part by the Intramural Research Program of the National Institutes of Health, National Institute of Environmental Health Sciences [ZIA-ES050165 to T.M.T.H.], and National Institutes of Health grants [R01NS114018 to Z.T.C., R35GM147091 to J.Z.]. Funding for open access charge: National Institute of Environmental Health Sciences. This work was supported in part by the United States National Science Foundation (MCB 2024964 to Jun Zhang).

## Author contributions

Protein expression, purification, and characterization: C.Q., R.N.W.; X-ray crystallography: C.Q.; Protein interaction and RNA binding analyses: C.Q.; Molecular Dynamics simulations: Z.Z., J.Z.; Writing and figure preparation: C.Q., Z.Z., R.N.W., Z.T.C., J.Z., T.M.T.H.; Project management: C.Q., Z.T.C., J.Z., T.M.T.H.

## Funding

## Competing interests

The authors declare no competing interests.
