## [Peer review file · Nature Communications]

REVIEWER COMMENTS

Reviewer #1 (Remarks to the Author):

Review of

Intra- and inter-molecular regulation by intrinsically-disordered regions governs PUF protein RNA binding

By Traci Tanaka Hall et al.

The manuscript describes an intrinsically disordered region (IDR) at the C-terminus of FBF-2 that autoinhibits its RNA-binding affinity. Moreover, there is a direct competition between the FBF-2 C-terminal region for interaction with the globular RNA-binding domain at the same site where LST-1 binds, both via "PIMs". This provides insight into the specific function of FBF-2 and differences with homologous Puf proteins. This clarifies data observed in genetic experiments that until now had not had a structural/molecular explanation. This will appeal to many in the fields of both genetics and structural biology.

The manuscript is beautifully written and very easy to read. Indeed, the only typo that I spotted was a missing full stop on line 279. Excellent!

All of the conclusions are completely reasonable given the data and methodology is sound. The only marginally troubling piece of data is the fits for the SPR data in Figure 5a. Particularly given what looks to be better fits in Fig 5b. Not a big deal, but maybe some comment could be made to that end? Methods are described in reasonable detail.

For Figure 6c I was slightly confused until I realised I had to start at the bottom of the panel and go back up toward a and b... Once I've read and and b, I'd rather carry on down or right to see the progression. Maybe just me?

Overall this is a rather well written and conceived manuscript. It could be published with very minor revisions.

Reviewer #2 (Remarks to the Author):

This manuscript uses structural and biochemical approaches to understand how partner proteins and intrinsically disordered regions (IDRs) impact RNA binding of FBF-2. The crystal structure of FBF-2 CT is novel and the proposed role of the CT in autoinhibition by intramolecular interaction with RBD is interesting. However, why this autoinhibition would be needed in vivo is less clear, especially in regard to the interaction with partner proteins. While the authors also mention this shortcoming, some attempt to address the biological consequences of the FBF-2 CT autoinhibition should be made to strengthen the impact of the structural/biochemical findings.

Points that need to be more clearly addressed regarding the conclusions in the manuscript are listed below:

1. In line 126, the authors first introduce the L610A mutation within the FBF-2 CT and state that the L610A substitution decreased the thermal stability of FBF-2 RBD+CT as indirect evidence of the CT with L610A mutation not interacting with the RBD. To demonstrate that the CT with L610 mutation no longer interacts with the RBD, in vitro assays to test binding of purified RBD and isolated CT (L610A) should be performed similar to what was done with the WT CT and RBD (Figure 2a).

2. The conclusions from Figure 6 are not clearly addressed in the results and experiments should be expanded to examine LST-1 dependence. Particularly as these represent major findings and are indicated in the abstract ("This regulatory mechanism, driven by IDRs, provides a molecular explanation for the interdependence of FBF-2 and LST-1 in germline stem cell self-renewal.").

Specific questions to address are: If FBF-2 RBD+CT Y479A, L610A, and Loop6A mutants show similar RNA binding affinities with wild-type FBF-2 RBD+CT with LST-1A or LST-1B, will those mutants still require LST-1 for germline stem cell (GSC) self-renewal in vivo? Do these mutants affect *gld-1* translation in vivo?

3. The broader impacts of IDR autoinhibition and partner protein interactions are thoroughly discussed in the Discussion and supporting data in Figure 7. As this is not a result, it is recommended that Figure 7 be moved to Supplemental Data. Alternatively, the authors could significantly improve the manuscript by testing any one of the potential hypotheses proposed about other PUFs—perhaps most importantly for their findings it the potential CT PIMs located in PUF-5, 6, 7 and FBF-1.

Other minor issues:

1. When discussing Figure 1c and 1d in the results section, the authors state that the additional density appeared near a loop in the “RBD” that connects repeats 7 and 8 (Line 91). Later, the authors mention that the FBF-2 “CT” (607-613) matched this new density (Lines 93/94). The text in this section gets somewhat confusing as to which amino acids are the R7-R8 loop and which are the CT ones interacting with the loop. It would help if Figure 1a indicates where R7-R8 loop is located within the domain structure.
2. Line 219, “Figure 3c, d” should be corrected to “Figure 2c, d”.
3. Line 140, the authors state FBF-2 RBD+CT had a nearly 5-fold “lower” affinity for *gld-1* FBEa RNA than FBF-2 RBD alone. In comparing K_d values for binding affinities, it should be stated that FBF-2 RBD+CT had a nearly 5-fold “weaker” affinity.
4. In Figure 1b, the function of the domains (“self renewal” and “spatial regulation”) should be removed as they are distracting and not the focus of the work/tested in this study.
5. Figure 3c and 3f are redundant and can be incorporated to other panels of Figure 3.

Reviewer #3 (Remarks to the Author):

Qiu, et al “Intra- and Inter-molecular regulation by intrinsically-disordered regions governs PUF protein RNA binding”

This is an excellent manuscript that describes an RNA binding protein in *C. elegans*, Fem-3 Binding Factor 2 (FBF-2) that regulates gene expression in stem cells and is part of the PUF family. FBF-2 has a well-ordered RNA binding domain (RBD) with 8 helical repeats that define the PUF RNA binding interface. This extended domain is flanked by intrinsically disordered segments at both ends. The focus here is on the C-terminal tail (C-Tail) which plays multiple roles. Embedded within this C-Tail is a small hydrophobic motif that the authors show is inhibitory to RNA binding. This is preceded by an acidic-rich region that the authors convincingly show can interfere with RNA binding by increasing the off-rate of RNA. The paper carefully shows how the C-Tail is inhibitory to RNA binding and then show how the Lateral Signaling Target 1 (LST-1) protein contributes to RNA binding by releasing the inhibitory potential of the C-Tail. The authors use a highly interdisciplinary approach to convincingly demonstrate the role of these motifs that are embedded in the C-Tail of FBF2.

The C-Tail, which is an intrinsically disordered region, contains a small motif that interacts with the RBD in the absence of LST, and this motif, dominated by a key leucine residue (L610), is shown in the structure that they solved. It is small and would not be recognized easily by sequence alignments. The authors convincingly show that this motif is inhibitory to RNA binding, and they use many different techniques to validate this hypothesis. This includes ITC to measure enhanced stability that is conveyed by the C-Tail and binding of the isolated C-Tail to the RBD which is abolished by mutating L610 and also by mutating Y479 which the structure shows is another key residue in this hydrophobic node that interacts with and stabilizes L610. They use ITC, SPR, and an electrophoretic mobility shift

assay (EMSA) to measure affinities and binding interactions. Their structure shows that the C-Tail contains a PUF Interacting Motif (PIM) that binds to the RBD in a similar way to the LST PIM motifs (LST-1A and LST-1B).

In contrast to the FBF-2 that lacks a C-Tail, the FBF-2+CT lowers RNA-binding affinity and does so by destabilizing protein-RNA interaction and accelerating dissociation of the RNA.

ITC showed that the melting temp was much less when CT was attached to the RBD. SPR shows that the Ct accelerates the off-rate of RNA. The authors then go on to show that LST-1, in particular, LST-1B, competes for the Ct of RBD, which allows it to then bind RNA with high affinity. Overall this is a rigorous analysis of an RNA binding motif that is regulated both positively and negatively by an intrinsically disordered region that can interact with other proteins as well as RNA. The family of PUF proteins is described nicely in discussion, and due to the non-conserved nature of the C-Tail, in contrast to the RBD, all are likely regulated in different ways to that this is a rich area to mine.

While acceptance is definitely recommended, there are a few questions that the authors could address.

1. The * band in the Figure 3b EMSA gel appears to be significant. Is this a valid "intermediate"?
2. The authors suggest that there may be some regulatory phosphorylation sites in FBF-2 or even potentially in LST-1. Are there any predicted P-Sites?
3. Are there any disease mutations in THE PUF family of proteins, especially in FBF-2, or in LST-proteins?
4. Are there other known binding partners for LST?
5. In the C-Tail, in addition to the electrostatic "Loop" and the small hydrophobic site, there is a linker that is highly conserved. IS there any indication what this segment does? It is likely doing something important given that it is so conserved.

Technical Suggestion: Figure 4. The authors should color the motif in yellow green so that it can be seen more clearly. The yellow is hard to see.

REVIEWER COMMENTS

Reviewer #1 (Remarks to the Author):

The manuscript describes an intrinsically disordered region (IDR) at the C-terminus of FBF-2 that autoinhibits its RNA-binding affinity. Moreover, there is a direct competition between the FBF-2 C-terminal region for interaction with the globular RNA-binding domain at the same site where LST-1 binds, both via “PIMs”. This provides insight into the specific function of FBF-2 and differences with homologous Puf proteins. This clarifies data observed in genetic experiments that until now had not had a structural/molecular explanation. This will appeal to many in the fields of both genetics and structural biology. The manuscript is beautifully written and very easy to read.

Indeed, the only typo that I spotted was a missing full stop on line 279. Excellent!
Thank you for the kind comments. We corrected the missing full stop.

All of the conclusions are completely reasonable given the data and methodology is sound. The only marginally troubling piece of data is the fits for the SPR data in Figure 5a. Particularly given what looks to be better fits in Fig 5b. Not a big deal, but maybe some comment could be made to that end? Methods are described in reasonable detail.

We agree that the fit of the curves to the data in Fig. 5b for FBF-2 RBD+CT looks better than the fit for Fig. 5a for FBF-2 RBD and the mutants in Suppl. Fig. 3. We could not improve the fit using the other two models in the instrument software for one-to-one binding, which account for target diffusion or ligand depletion. We contacted the Nicoya technical support about this, and they also were unable to achieve better fitting. Visual inspection of the curves seems consistent with the derived values. Importantly, the fit for deriving the k_{off} values from the dissociation phase is better, and the poor fit in the association phase should not affect the conclusions about the off rates for FBF-2 RBD vs RBD+CT. We now address this directly in the Methods section (page 15, lines 475-478).

For Figure 6c I was slightly confused until I realised I had to start at the bottom of the panel and go back up toward a and b... Once I've read a and b, I'd rather carry on down or right to see the progression. Maybe just me?

This is an excellent suggestion, and we appreciate the feedback. We rearranged Fig. 6c to move left to right with the addition of LST-1.

Overall this is a rather well written and conceived manuscript. It could be published with very minor revisions.

Reviewer #2 (Remarks to the Author):

This manuscript uses structural and biochemical approaches to understand how partner proteins and intrinsically disordered regions (IDRs) impact RNA binding of FBF-2. The crystal structure of FBF-2 CT is novel and the proposed role of the CT in autoinhibition by intramolecular interaction with RBD is interesting. However, why this autoinhibition would be needed in vivo is less clear, especially in regard to the interaction with partner proteins. While the authors also mention this shortcoming, some attempt to address the biological consequences of the FBF-2 CT autoinhibition should be made to strengthen the impact of the structural/biochemical findings.

Points that need to be more clearly addressed regarding the conclusions in the manuscript are listed below:

1. In line 126, the authors first introduce the L610A mutation within the FBF-2 CT and state that the L610A substitution decreased the thermal stability of FBF-2 RBD+CT as indirect evidence of the CT with L610A mutation not interacting with the RBD. To demonstrate that the CT with L610 mutation no longer interacts with the RBD, in vitro assays to test binding of purified RBD and isolated CT (L610A) should be performed similar to what was done with the WT CT and RBD (Figure 2a).

This is a good suggestion. We tested binding between the isolated CT peptide with L610A and the FBF-2 RBD by ITC. No binding was detected. These data are now presented in Fig. 2e and Suppl. Table 3.

2. The conclusions from Figure 6 are not clearly addressed in the results and experiments should be expanded to examine LST-1 dependence. Particularly as these represent major findings and are indicated in the abstract ("This regulatory mechanism, driven by IDRs, provides a molecular explanation for the interdependence of FBF-2 and LST-1 in germline stem cell self-renewal.").

Specific questions to address are: If FBF-2 RBD+CT Y479A, L610A, and Loop6A mutants show similar RNA binding affinities with wild-type FBF-2 RBD+CT with LST-1A or LST-1B, will those mutants still require LST-1 for germline stem cell (GSC) self-renewal in vivo? Do these mutants affect *gld-1* translation in vivo?

These are very interesting and important questions. We address some of them with published data, as summarized in #1 and #2 below.

1 - A published study (Wang, *et al.* 2016, ref. 41) generated worms that expressed truncated FBF-2, ending at G606 instead of D632, and therefore lacking the CT PIM. When this truncated FBF-2 was combined with an *fbf-1* loss of function background, ~11% of the progeny were sterile due to failure to initiate oogenesis contrasting with 0% sterility for WT FBF-2 in an *fbf-1* loss of function. This indicates a germline defect and supports our studies here. We present this information in the Results on page 4, lines 85-89.

2 - Animals expressing FBF-2 Y479A have been generated by Judith Kimble's lab, but the data are unpublished. This mutation disrupts interaction between FBF-2 and LST-1, but it also affects other partner protein interactions such as with SYGL-1, CPB-1, and GLD-3. A public seminar presented by Brian Carrick indicated that no defects were detected in germline stem cell self-renewal (see point #4 below), but there was a defect in the sperm-to-oocyte switch. This information has also been shared with us by personal communication with the Kimble lab.

3 - Judith Kimble's lab attempted multiple times to generate animals expressing FBF-2 L610A but were unsuccessful. This leads them to conclude that the mutation is dominant sterile. We identified the Loop6A mutant quite recently, so they did not try to generate this mutant.

4 - It is important to recognize that these animal experiments are quite difficult and time consuming. Germline phenotypes are not sufficient to understand the effects of the mutations in the animals. Although GLD-1 protein levels may be elevated in the distal germline, it does not necessarily result in sterility due to early commitment to meiosis (see Brenner and Schedl 2016, *Genetics*, 202:1085-1103). Therefore, the germlines must be probed for abnormalities in GLD-1 protein expression that are consistent across many animals, as described in Brenner and Schedl.

We agree that it would be fantastic to test the LST-1 dependence of these FBF-2 mutant proteins, but this will likely form the basis of an important future study. We changed the wording in the Abstract (page 2, line 25) and Introduction (page 4, line 78) to conclude more precisely: “This regulatory mechanism, driven by IDRs, provides a **biochemical and biophysical** explanation for the interdependence of FBF-2 and LST-1 in germline stem cell self-renewal.” This is what we meant by “molecular” but we understand how it could be misunderstood and we thank the reviewer for this comment.

3. The broader impacts of IDR autoinhibition and partner protein interactions are thoroughly discussed in the Discussion and supporting data in Figure 7. As this is not a result, it is recommended that Figure 7 be moved to Supplemental Data. Alternatively, the authors could significantly improve the manuscript by testing any one of the potential hypotheses proposed about other PUFs—perhaps most importantly for their findings it the potential CT PIMs located in PUF-5, 6, 7 and FBF-1.

We have moved Fig. 7 to Suppl. Fig. 4 in response to this suggestion, although we do feel the summary analysis is important enough to be a main figure. We absolutely agree that it is important to explore the potential CT PIMs in the other *C. elegans* PUF proteins. Experiments have begun to study FBF-1, but they will take time to complete. Testing the roles of potential PIMs in the other PUF proteins will require similar depth of experiments to what we include here. We feel that this manuscript has sufficient data for a thorough story.

Other minor issues:

1. When discussing Figure 1c and 1d in the results section, the authors state that the additional density appeared near a loop in the “RBD” that connects repeats 7 and 8 (Line 91). Later, the authors mention that the FBF-2 “CT” (607-613) matched this new density (Lines 93/94). The text in this section gets somewhat confusing as to which amino acids are the R7-R8 loop and which are the CT ones interacting with the loop. It would help if Figure 1a indicates where R7-R8 loop is located within the domain structure.

We have edited the text to directly include the residues in the FBF-2 R7-R8 loop (residues 476-489) when it is first mentioned (page 4, line 95). The loop in the RBD is labeled in Fig. 1d.

2. Line 219, “Figure 3c, d” should be corrected to “Figure 2c, d”.
Corrected (line 222).

3. Line 140, the authors state FBF-2 RBD+CT had a nearly 5-fold “lower” affinity for *gld-1* FBEa RNA than FBF-2 RBD alone. In comparing K_d values for binding affinities, it should be stated that FBF-2 RBD+CT had a nearly 5-fold “weaker” affinity.
Corrected (line 144).

4. In Figure 1b, the function of the domains (“self renewal” and “spatial regulation”) should be removed as they are distracting and not the focus of the work/tested in this study.

Done.

5. Figure 3c and 3f are redundant and can be incorporated to other panels of Figure 3.

Done.

Reviewer #3 (Remarks to the Author):

This is an excellent manuscript that describes an RNA binding protein in *C. elegans*, Fem-3 Binding Factor 2 (FBF-2) that regulates gene expression in stem cells and is part of the PUF family. FBF-2 has a well-ordered RNA binding domain (RBD) with 8 helical repeats that define the PUF RNA binding interface. This extended domain is flanked by intrinsically disordered segments at both ends. The focus here is on the C-terminal tail (C-Tail) which plays multiple roles. Embedded within this C-Tail is a small hydrophobic motif that the authors show is inhibitory to RNA binding. This is preceded by an acidic-rich region that the authors convincingly show can interfere with RNA binding by increasing the off-rate of RNA. The paper carefully shows how the C-Tail is inhibitory to RNA binding and then show how the Lateral Signaling Target 1 (LST-1) protein contributes to RNA binding by releasing the inhibitory potential of the C-Tail. The authors use a highly interdisciplinary approach to convincingly demonstrate the role of these motifs that are embedded in the C-Tail of FBF2.

The C-Tail, which is an intrinsically disordered region, contains a small motif that interacts with the RBD in the absence of LST, and this motif, dominated by a key leucine residue (L610), is shown in the structure that they solved. It is small and would not be recognized easily by sequence alignments. The authors convincingly show that this motif is inhibitory to RNA binding, and they use many different techniques to validate this hypothesis. This includes ITC to measure enhanced stability that is conveyed by the C-Tail and binding of the isolated C-Tail to the RBD which is abolished by mutating L610 and also by mutating Y479 which the structure shows is another key residue in this hydrophobic node that interacts with and stabilizes L610. They use ITC, SPR, and an electrophoretic mobility shift assay (EMSA) to measure affinities and binding interactions. Their structure shows that the C-Tail contains a PUF Interacting Motif (PIM) that binds to the RBD in a similar way to the LST PIM motifs (LST-1A and LST-1B).

In contrast to the FBF-2 that lacks a C-Tail, the FBF-2+CT lowers RNA-binding affinity and does so by destabilizing protein-RNA interaction and accelerating dissociation of the RNA. ITC showed that the melting temp was much less when CT was attached to the RBD. SPR shows that the Ct accelerates the off-rate of RNA. The authors then go on to show that LST-1, in particular, LST-1B, competes for the Ct of RBD, which allows it to then bind RNA with high affinity. Overall this is a rigorous analysis of an RNA binding motif that is regulated both positively and negatively by an intrinsically disordered region that can interact with other proteins as well as RNA. The family of PUF proteins is described nicely in discussion, and due to the non-conserved nature of the C-Tail, in contrast to the RBD, all are likely regulated in different ways to that this is a rich area to mine.

While acceptance is definitely recommended, there are a few questions that the authors could address.

1. The * band in the Figure 3b EMSA gel appears to be significant. Is this a valid “intermediate”? We have no evidence that the * band in Fig. 3b is a valid intermediate on the path to binding. EMSAs can have this type of extra band although the reasons are often unclear (as an example, see Fig. 1B in Stowell, et al., ref. 47). In this case, the band marked with an asterisk did catch our attention. To resolve whether it was non-specific or an “intermediate” and therefore part of the bound RNA, we conducted the SPR experiments. These SPR experiments confirmed the stronger affinity of FBF-2 RBD vs RBD+CT (and revealed the effect on off rate). Assigning this * band as part of the unbound RNA in the EMSAs results in a consistent conclusion.

2. The authors suggest that there may be some regulatory phosphorylation sites in FBF-2 or even potentially in LST-1. Are there any predicted P-Sites?

For the FBF-2 CT region, we used the program NetPhos3.1 to predict potential phosphorylation sites, and the serine residues are predicted to be phosphorylated. We now mention this in the text (page 11, line 310). For LST-1, NetPhos3.1 also predicts potential phosphorylation sites, some with scores >0.8. With little data on LST-1, we think it is best not to speculate on this now.

3. Are there any disease mutations in THE PUF family of proteins, especially in FBF-2, or in LST- proteins?

Humans express two PUF proteins, PUM1 and PUM2. Mammalian partner proteins are NANOS1, NANOS2, and NANOS3. At this point, our understanding of connections between mutant genes and human disease is limited but emerging. Most studies have made connections between misexpression of these proteins and spermatogenic failure (PMID: 233155541), leukemia (PMID: 36536516), colon cancer (PMID: 33508364), or retinoblastoma (PMID: 25100735), for example. One study identified mutations in PUM1 that are associated with ataxia, and another study showed that a point mutation in the 3'UTR of Dicer 1 disrupts PUM1 binding and results in degradation of Dicer1 mRNA and leads to cancer pre-disposition (PMID: 35736218). There are no LST-1 orthologs in mammals. Extrapolating our conclusions directly to studies of PUF protein partnerships in mammals remains difficult as little is known about the mammalian partnerships. The identification of partners that interact with mammalian PUFs to modulate their RNA-binding activity is an important line of investigation that we are keen to pursue and may result in an answer to the question that is raised here.

4. Are there other known binding partners for LST?

LST-1 has been shown to interact with the four PUF proteins in the 'PUF hub': FBF-1, FBF-2, PUF-3, and PUF-11 (page 10, line 282). Although LST-1 interactions have been identified in *C. elegans* protein interactome mapping, it does not appear that follow up studies have been performed to confirm the interactions.

5. In the C-Tail, in addition to the electrostatic "Loop" and the small hydrophobic site, there is a linker that is highly conserved. IS there any indication what this segment does? It is likely doing something important given that it is so conserved.

Indeed, the sequences of FBF-1 and FBF-2 have 91% identical residues, and the region between the end of the RBD and the beginning of the acidic region is well conserved, but we have not examined the function of this region. Thus far, the possible functions of four variable regions that are not conserved between FBF-1 and FBF-2 have been examined in worms by the Voronina lab (see Wang, et al., ref. 57). Their focus has been to identify why FBF-1 and FBF-2 knockouts have different phenotypes given that the high sequence conservation.

Technical Suggestion: Figure 4. The authors should color the motif in yellow green so that it can be seen more clearly. The yellow is hard to see.

We changed the color to make it a brighter yellow in the Pymol figure. We did not change to a yellow green because this is not in enough contrast with the blue and green of LST-1 A and B, respectively (see Fig. 1f). We want to maintain color consistency.

REVIEWERS' COMMENTS

Reviewer #2 (Remarks to the Author):

The authors have addressed all of my concerns

Reviewer #3 (Remarks to the Author):

The authors have adequately addressed the concerns that were raised by he reviewers. I recommend publication without further revisions.

Response to Referees

Reviewer #2 (Remarks to the Author):

The authors have addressed all of my concerns.

Reviewer #3 (Remarks to the Author):

The authors have adequately addressed the concerns that were raised by the reviewers. I recommend publication without further revisions.

Response:

We are grateful to the referees for their valuable input throughout the process.